# A Concise Review of *Dendrocalamus asper* and Related Bamboos: Germplasm Conservation, Propagation and Molecular Biology

**DOI:** 10.3390/plants10091897

**Published:** 2021-09-14

**Authors:** Anis Adilah Mustafa, Mohammad Rahmat Derise, Wilson Thau Lym Yong, Kenneth Francis Rodrigues

**Affiliations:** Biotechnology Research Institute, Universiti Malaysia Sabah, Kota Kinabalu 88400, Malaysia; anisadilah45@gmail.com (A.A.M.); mohammadrahmat93@gmail.com (M.R.D.); wilsonyg@ums.edu.my (W.T.L.Y.)

**Keywords:** bamboo, *Dendrocalamus asper*, micropropagation, plant tissue culture, DNA barcoding, genetic stability

## Abstract

Bamboos represent an emerging forest resource of economic significance and provide an avenue for sustainable development of forest resources. The development of the commercial bamboo industry is founded upon efficient molecular and technical approaches for the selection and rapid multiplication of elite germplasm for its subsequent propagation via commercial agro-forestry business enterprises. This review will delve into the micropropagation of *Dendrocalamus asper*, one of the most widely cultivated commercial varieties of bamboo, and will encompass the selection of germplasm, establishment of explants in vitro and micropropagation techniques. The currently available information pertaining to molecular biology, DNA barcoding and breeding, has been included, and potential areas for future research in the area of genetic engineering and gene regulation have been highlighted. This information will be of relevance to both commercial breeders and molecular biologists who have an interest in establishing bamboo as a crop of the future.

## 1. Introduction

Bamboo is the fastest-growing flowering perennial grass and considered as one of the world’s most important tree species [1]. Bamboos belong to the largest family of grasses, the Poaceae (Gramineae), and constitute the Bambusoideae subfamily [2]. With 121 genera and 1662 species [3], the bamboo population can be divided into three zones geographically: the American zone, the Asian Pacific zone and the African zone [4], and according to reference [5], about 80% of bamboo forest lands and species in the world are distributed across the Asian Pacific region. Bamboo, in general, plays an important role in human life, mainly in terms of meeting the current economic, ecological, and human essential needs [6,7]. Several studies have shown that bamboos cultivated commercially are more renewable and sustainable than other woody plants, as the inefficient harvesting and use of bamboo has become a major focus worldwide [8,9]. A current report by reference [10] stated that the global demand for bamboo is expected to reach a revenue of USD 98.3 billion with a Computed Annual Growth Rate (CAGR) of 5% by 2025. The same study predicted that the biomass energy market will reach USD 98.0 billion by 2027 with a CAGR of 9.2% from 2020 to 2027. Since bamboo is undeniably the most important non-woody forest resource in Malaysia and some Southeast Asian countries, the traditional timber industry needs to develop the use of non-timber bamboo into a value-added material such as floorboards, building materials, composite boards and furniture, as well as biomass products [11]. Therefore, this review aims to provide a detailed view of the biotechnology of *Dendrocalamus asper* and related bamboo species, as well as an updated data overview of most of the works published, especially in the areas of bamboo micropropagation techniques and the molecular identification of bamboos, to serve as reference material for the development of the bamboo industry.

## 2. *Dendrocalamus asper*

Taxonomically, *Dendrocalamus* belongs to the Bambuseae tribe and comprises about 35 species. In 2017, a study on *Dendrocalamus* and *Bambusa* conducted by reference [12] reported a higher similarity between these two genera when compared to other bamboo species. This finding supported the interpretation made by reference [13] that *Dendrocalamus* belongs to the same tribe as *Bambusa*. Besides, this similarity can be linked with their chromosome number, as with most species of the tropical bamboo genera, like *Bambusa*, *Cephalostychum*, *Dendrocalamus*, *Gigantochloa* and *Melocanna* reported to have a chromosome number of 72 (2n) [14]. *D. asper*, which is commonly referred to as sweet bamboo, is a multipurpose tropical clumping bamboo with high economic value [15,16]. Known also as rough bamboo, black bamboo or giant bamboo, *D. asper* grows to a height of 20–30 m, with a diameter of 8–20 cm and 20–45-cm-long internodes, and has relatively thick walls [17]. The origins of *D. asper* are not definitive, but according to reference [18], they are distributed across India and South East Asia, including Thailand, Vietnam, Malaysia, Indonesia and the Philippines. Recently, *D. asper* has been introduced in other tropical countries, including Ghana, Benin, DR Congo, Kenya and Madagascar. Figure 1 shows the distribution map of *D. asper* based on their endemic origin and subsequent introduction as an exotic species. Within tropical Asia, *D. asper* grows ideally in humid regions with rich, heavy soils, from lowlands to a 1500-m altitude, with an average annual rainfall of about 2400 mm. It can also survive well in semiarid environments with proper management. The mature stems are used to create furniture, musical instruments, household utensils, handicrafts and paper, while the upper internodes are used to make containers and cooking pots [19]. The tender young shoots are consumed as a vegetable and are thought to be the finest of all tropical Asiatic bamboos [20]. The rhizome, stems and branch cuttings can all be used to propagate *D. asper*. The propagules are grown in a nursery and then planted out in the field before or during the first half of the monsoon season after the roots have emerged. The best time to harvest stems is during the dry season; it is recommended to harvest mature stems that are 5–7 years old, while retaining some mature tillers in the clump.

Many excellent reviews on the efficient use and practical applications of bamboo can be found in the bamboo literature, with an emphasis on sustainable bamboo production in the sense of bio-economy and the circular economy [21]. This includes the review by reference [22], explaining the factors that emphasize bamboo growth and strength, as well as identifying ways to utilize bamboo in the industry, employment, climate change mitigation and soil erosion reduction. To date, only a few reports exist on the biotechnological applications for bamboo improvement, particularly related to industrial use. Representing studies on the in vitro tissue culture, references [23,24] focused on the progress in micropropagation of several important bamboo species via plant tissue culture techniques. Meanwhile, references [25,26] presented the review of the in vitro flowering of woody bamboo. In the review written by reference [27], it provided the limitations, progress and prospects of using biotechnological tools, as well as providing an overview of the molecular marker systems and their potential use in bamboo improvement. In their review on advances in bamboo molecular biology, the need for comprehensive bamboo phylogenetic research and the use of genomic knowledge and methods for bamboo breeding were underpinned by reference [28]. Further, references [2,29,30,31] overviewed the genetic diversity and characterization of bamboo by highlighting the use of molecular techniques in overcoming the problems associated with bamboo taxonomy and phylogeny. Besides, reference [32] overviewed the importance of DNA barcoding in bamboo identification, the ongoing research into the use of DNA markers in bamboo taxonomy and biosystematics, as well as the challenges encountered. The current path for bamboo improvement was highlighted by [33], culminating in a better understanding, conservation and improvement of bamboo for the future. The role of biotechnology [34], especially the areas of molecular genetics, genomics and quantitative genetics that can benefit bamboo production, have also been highlighted. In this review, we attempt to highlight the potential for bamboo development and conservation; more specifically, of *D. asper*, which is one of the most common commercially cultivated species.

## 3. Commercialization of *D. asper* in the Industry

### 3.1. Nutritional Composition 

Bamboo shoots are rich in proteins, dietary fibers, minerals and vitamins but are low in fat [35]. Secondary metabolites (phytochemicals) such as flavones, phenolic acids and steroids, which have antioxidant properties, are also abundant in the shoots [35,36]. Phytochemicals are known to possess many biological properties, which were discovered to be beneficial to human health [37,38], such as anticancer, antibacterial, anti-inflammatory [39,40] and antifungal activities [41]. A little-known fact of bamboo shoots is them being utilized as a food item. While most bamboo species produce edible shoots, only about 100 species have been reported to be used; these include *Dendrocalamus strictus*, *Dendrocalamus hamiltonii*, *Dendrocalamus giganteus*, *Dendrocalamus asper*, *Bambusa nutans*, *Bambusa bamboos*, *Bambusa tulda* and *Phyllostachys pubescens* (Moso) [20,42]. In China, *P. pubescens* (Moso) has been largely cultivated and utilized for its shoots. In India, *D. strictus* and *B. bamboos* are the most common species used as food. *D. asper* is an important edible bamboo species in Thailand []. The sweet bamboo shoots are primarily used fresh, dried, shredded, pickled, canned or fermented [43].

Though edible bamboo shoots are high in nutrients, a lack of information on the nutritional composition is a limiting factor [41]. The nutritional compositions, biological activities and phytochemical contents of the shoots of *D. asper* collected from different regions of Malaysia, Peninsular (Perak) and East Malaysia (Tambunan and Sabah) were reported by reference [41]. Within these two regions, the potential of *D. asper* shoots as a source of nutrition was highlighted in their work. Important elements such as iron, zinc, manganese, copper and selenium from the fresh juvenile shoots of India’s *D. asper* were reported by reference [44]. As stated by reference [45], these minerals are constituents or cofactors of several essential enzymes involved in the biological functions of body cells. Furthermore, the nutritive properties found in these juvenile bamboo shoots have potential commercial applications [20]. In addition, reference [46] discovered a new compound named 4-hydroxybenzoic acid in *D. asper* shoots that has been reported to have antiepileptic effects in Chinese traditional medicine. This shows that *D. asper* has the potential to be cultivated for their edible bamboo shoot for medicinal use. 

### 3.2. Renewable Construction 

The production of wood from natural forests has been declining in recent years. Concurrently, the global population has been growing, especially in Asia and Latin America, driving up the demand for high-quality wood for construction [47]. As a result, the need for non-wood resources as a replacement for wood has increased. Bamboos have been traditionally used as a construction material for rural housing and food, as well as handicraft products [33], and the use of bamboo has been explored recently for construction in a variety of engineering applications, such as housing, bridges and flooring. Besides, most studies on their mechanical properties have shown that bamboo is one of the renewable, sustainable materials with a biomass yield per hectare comparable to tropical forest hardwood [48]. Due to its sustainability, flexibility and mechanical strength, it is comparable to teak wood and mild steel [49]. However, despite its status as a natural engineering material, bamboo has not been thoroughly explored in many Asian countries as a source of renewable construction material. Recently, there has been increased interest in the design and application of structural Bamboo Products (SBP) using Moso (*P. edulis*), Guadua (*G. angustifolia*) and Tre Gai (*B. blumeana*) bamboo [50]. 

*D. asper* has been studied for the production of composite lumber and has been discovered to be a high-quality alternative to wood, because it has a similar pH value as compared to the wood species used in construction [47]. The values pH of *D. asper* for the bottom, middle and top sections of the culm show no variations (5.36, 5.45 and 5.38, respectively). This acidic value is very important for composite manufacturing, as, for species with higher pH levels, an additional catalyst is needed during hot pressing. Moreover, with the influence of a salt treatment, the Philippines determined that *D. asper* has excellent mechanical properties for construction building materials by highly influencing its compressive and bending strength [51]. Besides, a test on the differences of bamboo compressive strength and tensile strength showed that *D. asper* possess excellent mechanical properties in the compression and tensile strength, which indicate a good quality for construction material [52]. This demonstrated that, following a treatment with traditional preservatives, bamboo can expand its usable lifespan and has sufficient durability for long-term structural applications. 

### 3.3. Renewable Energy 

The sustainability and diversification of energy supplies are critical challenges for global energy sectors. The concern towards meeting a sustainable energy supply has resulted in a rise in the renewable energy demand. Biomass energy is the most commonly used form of renewable energy in the world, and in this context, bamboo is one alternative that stands out, as it has a great deal of potential for producing energy [53]. Bamboo has a satisfactory calorific value when compared to other plants, and its biomass can be used both in the raw form (in natura) and can be processed in the form of charcoal, briquette, pellet, syngas and biofuel [54]. Bamboo also produces more biomass than other energy crops, like poplar, switchgrass, miscanthus, common reed and bagasse [55,56]. Malaysia still relies on nonrenewable resources as its primary source of energy, with all major power plants still generating electricity using fossil fuels such as oil, gas and coal [57]. According to reference [58], the global energy demand, which was 540 quadrillion British thermal units (Btu) in 2012, is expected to rise to 815 quadrillion Btu by 2040, including in Malaysia, and is expected to experience an accelerated economic growth. Though China and Japan have conducted the majority of scientific research on bamboo as a biofuel feedstock, there has been little study on tropical bamboo species [59]. *D. asper* is one of the bamboo candidates with the characteristics and ability to be used as a pelleted energy source [60]. Moreover, reference [61] stated that *D. asper* can be used to manufacture briquettes at 80 °C without losing its efficiency and quality, as well as lowering energy costs. Reference [62] presented *D. asper* as having the energy characteristics that suggest its ability for use as an energy source in the form of charcoal. Moreover, *D. asper* has been shown to be ideal for generating electricity due to its unique properties and appropriate growth rate [63]. From all the studies presented, the bamboo of *D. asper* has the potential to be recognized for creating sustainable gains from the industrial sectors.

## 4. Bamboo Propagation and Diseases

### 4.1. Traditional Propagation of Bamboos and Tissue Culture

Conventionally, bamboos are propagated through seeds. Short seed viability periods of 3–6 months, long-term gregarious flowering, the monocarpic nature of the plant, poor seed set and large-scale seed consumption by pests are all factors that restrict the use of seeds as a reliable resource of propagation [23,64]. Owing to the segregation of their traits, the genetic homogeneity of seed-based progenies is also in question. As a consequence, vegetative propagation from layering, off-set and rhizome planting, marcotting and branch and culm cuttings are used for propagating the bamboos [17,65,66]. The traditional bamboo propagation methods, on the other hand, are detrimental to mother plants during collection, involving high labor costs, transportation difficulties, bulky materials and seasonal dependence, which is typically limited to a short period of time, and these techniques are only effective for small-scale production [67,68]. The first report on a successful tissue culture of bamboo was done by reference [69], who described the embryo culture of *D. strictus* on a sucrose-enriched medium. In vitro propagation provides the ability to acquire large progenies from elite genotypes, since it was believed that it could solve most of the problems associated with conventional propagation [70]. In most cases, when designing protocols for in vitro plant propagation, trial-and-error experiments are needed to identify specific conditions for individual species, genotypes and even the donor plant development stages [25]. The aim of bamboo tissue culture regeneration protocols is to achieve the large-scale production of plants for operational planting, to produce disease-free and genetically uniform planting material and to provide material for breeding programs, as well as germplasm conservation [71]. 

### 4.2. Bamboo Diseases

Tissue culture has become a platform to confer resistance against specific diseases by manipulating the genetics of the plant systems. Bamboo tissue is composed primarily of Hemicellulose, Cellulose and Lignin [72], and microbes that rely on these as a source of carbon represent potential pathogens. Investigations carried out in China [73] in 148 bamboo species over a 11-year period from 1995 to 2006 recorded 208 potential pathogens, the majority of which were fungi (108). Similar studies carried out in Japan [74] reported 257 fungal strains, of which 75 could be identified using 18S rRNA gene and internal transcribed spacer region (ITS) analyses with Xylariales as the major dominant group. Bamboo died back, which was caused by the fungus *Aciculosporium take*, reported to predominantly infect *Phyllostachys bambusoides*, with a lower incidence in *Phyllostachys pubescens* Western Japan [75], leading to phenotypic changes referred to as the ”witches’ broom” of bamboo. Among the pathogens reported from India [76], *Bambusa nutan* was found to be infected by *Nigrospora oryzae*, the causative agent of leaf spot disease, whereas *Fusarium oxysporum* and *F. verticillioides* were dominant on *Bambusa balcooa* and *D. asper*, respectively. Another extensive study carried out in India across 12 phyla and 46 orders identified the pathogens belonging to the phylum Deuteromycota and Ascomycota as causative agents for foliage-related diseases. Basidiomycota was found to be associated with culm diseases, which is supported by the evidence that white rot fungi belong to this phylum and are involved actively in the degradation of lignin [77] and the utilization of carbohydrate complexes [78] that constitute the structural elements of bamboo. Interestingly, not all microbiota associated with bamboos have been reported to be pathogenic, with reports providing evidence of endophytic bacterial communities [79] associated with the rhizomes of tropical bamboos that share a unique symbiotic relationship and may also serve as a means of host defense [80] and biological control. Recent reports of the new fungal pathogen *Arthrinium bambusicola* in Thailand [81] and novel techniques such as high-throughput genome sequencing have provided new tools for the discovery and diagnosis of fungal pathogens [82], the early detection of which is critical to their control. 

The Bamboo Mosaic Virus (BaMV) is one of the most well-documented and studied among the viruses associated with bamboos [83], although many individuals may be asymptomatic carriers with no physical evidence of viral infection. The mode of transmission of the virus appears to be mechanical injury, either via routine farming operations or insect vectors [84]. Recent reports have emerged of etic recombination events in Indonesia [85], which indicate Taiwan as the origin of the virus. Molecular data has provided evidence of the factors involved in the intracellular movement of the virus, which is mediated by movement proteins [86], and measurements for the containment and eradication of the BaMV have included treatments with abscisic acid [87], which has been reported to induce resistance and improve the host defense, as well as the application of interfering satellite RNA [88] in transgenic bamboo plants. The adoption of pertinent biosecurity measures during import of the germplasm, as well as the monitoring of invasive pathogens in commercial plantations, is currently the best available measure for the control of BaMV and other pathogens. 

## 5. Regeneration of *D. asper*

In any plant tissue culture, choosing the appropriate propagation method is crucial. Different routes such as direct shoot induction (axillary shoot proliferation), the production of adventitious buds through organogenesis and somatic embryogenesis are pathways of choice for the rapid and large-scale propagation of bamboo using both juvenile and mature explants [89]. Reference [90] stated that callus have three basic developmental ways: somatic embryo development, shoot organ differentiation and a mixed development pathway that includes both somatic embryogenesis and shoot organogenesis. A developed in vitro culture of *D. asper* was successfully established from various explants. Seeds [91], mature plants of nodal explants [16,19,91,92,93,94,95,96], stem cuttings [97], small branch cuttings [98], nodal and leaf bases [99], inter node segments [100] and clump [101] were successfully used for the mass propagation of *D. asper* in vitro. Some researchers also used inflorescence explants for establishing protocols for the multiple shoot proliferation in *D. asper* [102] and in vitro flowering studies [103]. Most of the research available in *D. asper* used juvenile and mature tissues on enhanced axillary branching, with just a few reports on indirect organogenesis. By using nodal explants, the first regeneration of shoots and roots from *D. asper* callus was carried out by reference [92]. Moreover, several authors published studies incorporating both in vitro and in vivo multiplications of *D. asper* to improve the multiplication rate and measure the quality of plantlets in the field. Table 1 and Table 2 below respectively show the in vitro regeneration of *D. asper* through organogenesis and somatic embryogenesis using different explants. According to reference [25], because of the size and diversity of this plant family, establishing the best culture medium, combinations of plant growth regulators and other compounds in promoting the growth of explants will usually take several months. Therefore, to reduce the current gap between demand and supply, cost-effective methods for planting large-scale bamboo propagation in new bamboo forests must be established.

### 5.1. Explant Surface Sterilization

Explant contamination is determined by several factors, including the source of the explants and the ambient environment [113]. The presence of latent infection was found to be one of the major constraints in achieving a contamination-free bamboo culture in vitro. A high incidence of surface and systemic contamination is one of the primary reasons for the failure of bamboo tissue cultures, particularly in mature bamboo [114]. Large intercellular spaces and vessel cavities at the cut ends of single-node explants from lateral branches, which are commonly used in bamboo tissue cultures, provide a space for contaminating agents that remain within the tissues during sterilization treatments. The presence of these microorganisms causes plant mortality, growth variation (reduced shoot proliferation and rooting) and tissue necrosis [115]. Therefore, many sterilization treatments are ineffective and only postpone the contamination from spreading at the early stages [116]. Several researchers have documented alternative methods for controlling contaminants, such as using light and heat [117], as well as microwave and hot water treatment [118]. Obtaining contaminant-free bamboo explants has been difficult in some cases [119]. Most researchers have relied on chemicals to control contaminants during plant in vitro propagation, and just a few papers focused on explant disinfection [120,121]. Surface sterilization is a vital step in preparing healthy and viable explants in tissue cultures [122,123]. The type of sterilant use, its concentration and contact time vary for different plant tissue cultures [124]. In bamboo species, there are only a few disinfection methods that are successful with a combination of sterilant, antibiotic and fungicide. Therefore, it is necessary to employ different and sequential protocols containing antimicrobial agents to solve this problem. Table 3 shows the list of sterilants used to treat different explants in *D. asper* disinfection.

Mercuric chloride (HgCl_2_), sodium hypochlorite (NaOCl) and potassium hypochlorite (KClO) or calcium hypochlorite (CaOCl), hydrogen peroxide (H_2_O_2_) and ethanol (C_2_H_5_OH) are the most widely used surface sterilants in bamboo micropropagation [115]. Among them, mercuric chloride (HgCl_2_) proved to be the best sterilant for nodal explants, while sodium hypochlorite (NaOCl) is preferred by some other authors [119]. It is suggested that 0.1% HgCl_2_ was the most efficient for nodal bamboo disinfection, since a high concentration of HgCl_2_ could minimize the plant’s growth due to the chemical’s impact on the internal tissues [125], while the surface sterilization of seeds is usually accomplished by treating them with 0.05–0.1% HgCl_2_ for 15–20 min [126]. According to reference [127], when 0.2% HgCl_2_ is used to treat *B. wamin*, it reduces the plant’s bud breaking percentage, while sterilizing the *D. asper* nodal segments with 0.1% HgCl_2_ increased the chance of an aseptic culture [16,19,91,93]. Reference [128] reported a similar result in the nodal explants of *Aconitum heterophyllum*, finding that 0.1% HgCl_2_ was superior to 10% H_2_O_2_ and 1.5% NaOCl. In addition, reference [129] sterilized *B. tulda* and *M. baccifera* with 0.1% HgCl_2_ and resulted in a low rate of contamination. References [130,131] found that treating explants with 0.1% HgCl_2_ for a shorter period of time enhanced the explants’ survivability and bud break frequency. Similarly, a lower duration of 5 min with 0.1% HgCl_2_ increased the survival rate of *D. hamiltonii* Arn. Ex Munro [132]. The use of NaOCl as a sterilant was also explored by different researchers for surface sterilization in bamboo species [133,134]. For instance, reference [135] used 0.1% NaOCl to sterilize axillary buds in the dark for 24 h, followed with NaOCl containing 2.5% chlorine for 15 min. Twenty percent of Domestos containing 4% NaOCl for 30 min was used in sterilizing *D. hamiltonii* [136], and 4% NaOCl was also employed by reference [106] for the surface sterilization of *D. asper* seeds. 

Chemical penetration during surface sterilization, especially at the cutting ends, has a toxic effect and slows down the plant growth [116]. This entire chemical effectively eliminates the surface contamination, but it has limitations in the ability to control endophytic contamination. Different researchers have used antibiotics like kanamycin, ampicillin sodium salt, Savlon, Tempol, streptomycin sulphate, gentamicin, cetavlon, tetracycline hydrochloride, bacteriocin and so on to reduce the spread of contamination [137,138,139,140]. In *Dendrocalamus* bamboo, most of the researchers used streptomycin sulphate as the main antibacterial agent in the surface sterilization process [19,25,94,120,132,141]. In addition, reference [142] reported that the antibiotics streptomycin, rifampicin and ciprofloxacin were effective in disinfecting nodal explants of *B. tulda*, *B. balcooa*, *B. bamboos* and *D. asper*. Bavistin, Benomyl and Mancozeb are the most commonly used fungicides in bamboo micropropagation. Despite the fact that Bavistin and Benomyl contain the same active ingredients, Benomyl is the most effective chemical for controlling fungal contamination [143]. However, in comparison to Bavistin, the use of Benomyl for bamboo in vitro propagation is very limited, including *D. giganteus* and *B. vulgaris* [144], *D. giganteus* Munro [145], *B. vulgaris* “Striata” [146] and *G. angustifolia* [147].

For Bavistin, different concentrations were tested on bamboos: 0.1% for *B. nutans* Wall [121], 0.5% in *B. nutans* [148] and 0.2% in *D. strictus* [149]. However, during in vitro bamboo propagation, the most commonly used concentration of surface sterilants was reported to be 1% [16,150,151,152]. As for *D. asper*, the most common concentration used from Bavistin is 0.2% [19,93,110]. Antibiotics are effective against different bacterial contaminants, but they may be harmful to plants and may lead to the selection of resistant bacterial strains after prolonged exposure [153]. Therefore, the development of new compounds and formulations have been created for decreasing the rate of contamination, namely the Plant Preservative Mixture^TM^ (PPM^TM^) and Vitrofural [154]. According to reference [147], 2.0% of PPM^TM^ effectively reduced the contamination on *G. angustilodia*, while Vitrofural, an antimicrobial agent, was also effective in reducing the occurrence of microorganism contamination when added to the culture medium of *D. asper* [101]. The efficiency of these formulations in controlling microorganisms was said to be dependent on the genotype of the microorganism, the concentration of the sample and the time after the assay started.

### 5.2. Organogenesis

Organogenesis is a two-step process that involves shoot proliferation or induction on explants, followed by root meristem induction [27]. When shoot tips or, in the nodal, shoot segments are used as explants, the dormant axillary bud is released from apical dominance during in vitro propagation. The removal of apical dominance for the enhanced axillary branching is required in order to achieve the de-repression and multiplication of shoots, and it has become a reliable technique due to its ease of use and rapid propagation rate [155]. Plants regenerated from shoot tips or nodal buds involve the normal ontogenic route for branch development, considered to be genetically stable and free of the somaclonal variations present in callus-derived plants [25]. Besides, axillary shoot proliferation has proven to be the most practical and reliable approach for large-scale production [119,156,157]. 

The success of the regeneration pathways depends on several factors, including plant growth regulators that regulate the dedifferentiation and redifferentiation of plant cells and the choices of explants, as well as their genotypes. Details on the level and kind of plant growth regulators (PGRs) added in the culture medium largely determine the success of tissue culture protocols. As mentioned by reference [158], exogenous hormones are the most important factor in achieving the optimal level of plant multiplication. The incorporation of BAP into the medium was found to be the most effective for the bud breaking and shoot proliferation of several bamboo species. These findings are consistent with the previous research on the in vitro propagation of various bamboo species, such as *Guadua angustifolia* [147], *Dendrocalamus giganteus* [159], *Guadua magna* [160], *Bambusa glaucescens* [161], *Bambusa balcooa* [162],*Gigantochloa atroviolaceae* [150] and *Dendrocalamus hamiltonii* [91], in which BAP are widely used and proven to be effective. In intact plants, high levels of cytokinins have been shown to adversely affect growth by promoting programmed cell death in cell cultures, the necrosis of leaves, reduced root growth and promote ethylene biosynthesis. Although the use of cytokinins results in maximum shoot formations, at a certain concentration, abnormal forms on the shoots produced are induced [163]. The levels of cytokinins used in the medium can be seen in reference [106], aprevious work using *D. asper* seeds where increased BA levels (7–10 mg/L) resulted in 25–30 shoots per seed. With the increased BA concentration (1–10 mg/L) in the medium, however, the shoot length decreased by 4 cm to 0.5 cm. Similarly, a higher concentration of BAP reduced the rate of shoot multiplication and the rate of bud break from *D. asper* axillary buds in the study carried out by reference [101]. Using the same species, reference [19] observed a higher rate of bud sprouting (90%) in response to 15-µM BAP, and this value decreased at higher concentrations. On the other hand, reference [94] showed (90%) nodal sprouting in *D. asper* cultivated with 8.86-µM BAP and declined with an increase of 13.29-µM BAP. These results are also supported by the previous studies of reference [159] in *D. giganteus*, reference [150] in *G. atroviolaceae* and reference [16] in *D. asper*. As a result, the BAP concentration in the medium had a remarkable impact on the in vitro bamboo shoot-forming capacity. Figure 2 depicts the shoots multiplication of *D. asper* in 3-mg/L BAP cultured from a seeds explant.

For shoot multiplication, the synergistic effect of two cytokinins, Kn and BA, was found to be the most effective in *D. giganteus* [159] and *B. glaucescens* [161]. Reference [129] obtained better shoot multiplication, as well as a better quality of shoot texture, in both *B. nutans* and *M. baccifera* culture initiation by combining cytokinins. When Kn are used alone in a media, a lower rate of explant establishment and lesser number of shoots were observed compared to BAP [19]. This statement was highly supported when BAP alone was found to be more effective than Kn in inducing axillary shoots in *D. asper* [16,19,93,110]. Similar results were also found on a different bamboo species, *G. atroviolaceae* [150]. Besides, the use of Adenine Sulphate (AdS) additives in the earlier studies also resulted in axillary shoot proliferation to a few bamboo species, such as *D. strictus* [164] and *B. vulgaris* [165]. Since adenine has a close base structure to cytokinins and has cytokinin-like activity, supplementing the medium with AdS can stimulate cell growth and enhance shoot formation According to reference [166], in a combination or synergism with endogenous or exogenously supplemented PGRs, the addition of AdS acted as an elicitor or enhancer for cell growth. Reference [16] demonstrated that adding AdS to the media increased the number of *D. asper* shoots and their proliferation. Similarly, in the same species, reference [19] found that adjuvating the media with AdS resulted in a substantial increase in the shoot multiplication rate. Thidiazurom (TDZ) has also been applied to regulate the proliferation of axillary shoots in *B. oldhamii* [167], *D. strictus* [168] and *Costus speciosus* [169]. As stated by reference [170], TDZ reduces the senescence, resulting in an increase in the explant biomass. In a tissue culture of *D. asper* nodes, media supplemented with TDZ was able to achieve high percentages of shoot formation, and according to reference [96], adding TDZ to the media resulted in the highest percentage of *D. asper* shoots (80%), while using it in a lower concentration resulted in more leaves than the other cytokinins. Similarly, these results were also supported by reference [135] for *D. hamiltonii*. Furthermore, Meta-topolin 6-(3-hydroxybenzyl amino) purine (mT), a recently developed form of aromatic cytokinin, has been used to enhance the micropropagation protocols of other plants, primarily to minimize the physiological abnormalities, diminish the rooting inhibitory effect characteristic of Cytokinins and, thus, to improve rooting in the acclimatization process [171]. As a result of the addition of mT to *D. asper*, reference [101] observed an increase in the culm proliferation.

Apart from the optimum concentration of growth regulators, the selection of the appropriate number and size of shoot propagules was also crucial for adventitious shoot multiplication. Under in vitro conditions, single shoots often do not survive. According to references [91,106], the shoot multiplication rate declined sharply if propagules of less than three shoots were cultured in *D. asper*. These findings were also supported by references [94,172], which found that shoot clumps rather than single shoots are better for bamboo multiplication through axillary shoot proliferation. References [23,91] specified that, in *D. asper*, a propagule of two to three should be chosen from healthy growing multiple shoots ranging in length from 1.0 to 2.0 cm long for the best root induction. Furthermore, the rooting percentage was found to be lower in longer shoots (>2.0 cm) with folded leaf lamina. In contrast, however, reference [19] found that eight shoots per propagule were the most effective for *D. asper* shoot multiplication. In different species, reference [173] stated that plantlets associated with two or three shoots are more suitable for greater new shoots and roots in *Guadua aff.* Chaparensis. Additionally, a propagule of three to five shoots has been reported best for the multiplication of shoots in *B. tulda* [174], *D. longispathus* [175] and *D. latiflorus* Munro [176]. Meanwhile, reference [177] reported that a 20-fold shoot multiplication rate with a propagule size of three to five shoots in *D. hamiltonii* and a propagule of 7–10 shoots was found as the optimum (supporting a five to six-fold multiplication rate) for the large-scale propagation of *D. hamiltonii* [19,132]. 

### 5.3. Somatic Embryogenesis

Micropropagation via somatic embryogenesis is another simple and effective method for mass propagation, since both the root and the shoot primordia are produced in a single step [27]. Somatic embryogenesis has the ability to produce the greatest number of plantlets in the shortest amount of time, allowing for the use of bioreactors for large-scale somatic embryo production and delivery through artificial seed encapsulation [178]. Somatic embryogenesis is the morphological differentiation of somatic cells into somatic embryos under the control of growth regulators. Explant selection is the most significant element in obtaining somatic embryogenic callus. Seeds, seedlings, isolated embryos, roots, shoot apices, leaf explants, internodes and anthers have all been used to raise Calli. Immature embryos can also be used to induce embryogenesis. In most cases, this capacity is genetically regulated, meaning that individual genotypes within a species can vary in their ability to undergo somatic embryogenesis [179]. However, this somatic embryogenesis, on the other hand, can cause somaclonal variations due to the physical and morphological changes in the undifferentiated callus [180]. As a result, most studies discussed the micropropagation of bamboo using enhanced axillary branching, some directly from seeds, but only a few reports documented indirect organogenesis. 

2,4-D was commonly used to induce the formation of callus via somatic embryogenesis, depending on their optimal concentrations and the bamboo species used [181]. For example, references [95,99] added 20–30-μM 2,4-D in the regeneration of *D. asper* to induce callus, while reference [112] used 3-mg/L 2,4-D, and both showed the successful formation of callus. However, adding a higher concentration of 2,4-D can cause a detrimental effect for the culture. An earlier study conducted by reference [182] used 2,4-D alone at higher concentrations to yield friable callus, which was unresponsive for *Dendrocalamus*. Furthermore, the synthetic auxin 2,4-D has been linked to genetic defects such as polyploidy and DNS synthesis stimulation, which may lead to mutations [183]. As for organogenesis via callus, it usually takes place when 2,4-D is used in combination with BAP, IAA, NAA and IBA [92]. The combination of PGRs could promote callus differentiation, shoot elongation in the regeneration of wood and herbaceous bamboo species [184,185,186]. According to reference [187], the induction of somatic embryogenesis for *D. hamiltonii* only occurred in the nodular and compact callus by gradually decreasing the concentration of 2,4-D and NAA in the medium, with a corresponding increase in BA concentration. On the other hand, reference [92] observed that the combination of 1-mg/L 2,4-D and BAP is the most suitable for inducing both shoots and roots from the callus of *D. asper*, while reference [112] observed that BA did play an important role in callus differentiation of *D. asper*, and the medium containing both BA and KT showed significantly higher callus differentiation than the medium containing only BA.

### 5.4. Rooting

The most important steps in any micropropagation procedure are inducing roots from dissected shoots and the survivability of plantlets in the soil. In general, IBA, NAA and IAA are the common auxin for rooting in tissue culture, and IBA has been found to be the most commonly used auxin for the in vitro rooting of bamboo shoots. Some studies show that cytokinins (BAP or Kn) in combination with auxins were also used to induce root growth in bamboos. When propagules of *D. hamiltonii* were cultured on IBA-supplemented media alone, reference [188] reported a very high rooting efficiency (100%). Similar findings on the stimulatory effect of IBA on *D. asper* rooting was reported by another researcher [16,93,106]. Reference [16] reported that IBA was better than IAA, as IAA was found to be ineffective in the root induction and elongation of *D. asper*. The result was supported by an earlier study by reference [175] on *D. longispathus*. NAA treatments were also favorable for enhancing rooting in *D. asper* mature shoots tissue [102,106] and other important bamboo species [97,189]. Moreover, in both *D. asper* and *D. membranaceus*, a combination of NAA and IBA was found to be the most effective in rhizome induction [109]. Therefore, the physiological activities of various auxins vary based on how well they pass through tissues, stay bound within cells or are metabolized [19].

### 5.5. Hardening and Acclimatization

One of the most important steps in the field transfer of hardened micro-propagated plants is transferring them to an ex vitro environment, and their survivability performance in the field. This is due to the changes experienced by the in vitro-raised plantlets when they are moved from a high humidity condition to a natural environment with a higher irradiance and low humidity. Although in vitro raised plants have well-developed roots, they typically have leaves with weak or no development of cuticular wax, impaired stomatal mechanisms and low photosynthetic pigments, which prevent the plant from directly surviving the in vivo environment [25]. Therefore, an efficient hardening and acclimatization technique is needed to ensure that in vitro-raised plantlets survive in the field. In general, healthy and well-rooted plantlets are washed to remove the rooting medium and transferred to a pot containing growth-promoting compositions, such as soil, sand, perlite, soil rite, agro peat, cocopeat, compost, vermiculite, farmyard manure and so on, either alone or in different ratios and keep them under high humidity [19,190]. The majority of studies have used the described substrate in a 1:1:1 ratio or modified. Reference [93] reported that the propagation of *D. asper* resulted in a 95% survival rate after 30 days using the 1:1:1 ratio of sand:soil:farmyard manure, and reference [16] resulted in 98% well-acclimatized *D. asper* plantlets with the same ratio of substrate. Furthermore, references [19,132] used the same mix combination and successfully transferred 25,000 *D. asper* and 3000 *D. hamiltonii* plants to the Forest Department land in Yamunanagar after reporting a success rate of 92.34% and 100% for *D. asper* and *D. hamiltonii* in the greenhouse, while 79.76% and 85% success were achieved in the field, respectively. Meanwhile, different combinations of soil-hardening mixtures have been identified by several workers; soil:sand:compost cocopeat (1:1:1) in *B. nutans* [191], soil:sand:cow dung (1:1:1) in *B. Pallida* [148], cocopeat:vermicompost (3:1) in *B. balcooa* [149] and sand:soil:peat moss (1:1:1) in *D. latiflorus* [176].

In a few cases, researchers have discovered that applying reduced MS minerals to plantlets is essential for better acclimatization. This application was done in hardening *D. asper* [192] and *D. hamiltonii* [193] by transferring in vitro plantlets to a half-strength MS liquid medium without plant growth regulators and vitamins. As conducted by reference [106] on *D. asper*, the developed root plants were hardened in soilrite and fed with half-strength macro- and micro-MS medium, and a 95% survival rate was obtained with 6000 plants from six individual seed lines formed in soil. With the use of dune sand and vermicompost (3:1) and irrigated with 1/4x MS liquid medium without sucrose, reference [19] obtained a 90–95% survival of *D. asper* plantlets. However, according to reference [147], some bamboo species demonstrated a direct acclimatization of in vitro-raised plantlets under greenhouse conditions without a hardening phase. For example, the *B. bamboos* in vitro plantlet was successfully acclimatized with soil without the use of other substrates [137]. However, because of their inability to tolerate biotic and abiotic stresses, the mortality rate of in vitro-propagated plantlets increases as they are directly transferred to an external environment [126]. Therefore, maintaining healthy rooted plantlets in a potting mix with a 1:1:1 composition of sand:farmyard manure:soil is the most frequent procedure for *D. asper* before transferring them to a greenhouse [25,194]. 

Macro-proliferation is a method of plant multiplication that involves separating the rooted tillers to improve the rate of multiplication of in vitro-raised plants and to ensure a continuous supply of plantlets [27]. In general, this approach is appropriate for species that produce seeds. After successful acclimatization, macro-proliferation can be very suitably adapted for well-established bamboo plantlets after three to five months of transfer of the micro-plants [23]. This technique was successfully used by references [195,196,197]. By adapting the macro-proliferation after the micropropagation of in vitro-raised plants, the splitting of rooted tillers has been proven in doubling the production of *D. asper* plants [19], while a three-fold increase was achieved in *B. tulda* [190] and *B. balcooa* [162]. 

## 6. Bamboo Dormancy and Bud Breaking

### 6.1. Seed Dormancy

The word “dormancy” refers to the temporary stop of plant growth. It comprises true dormancy, known as (“rest” or “endodormancy”) triggered by internal factors and climatic dormancy (“quiescence” or “ecodormancy”) controlled by external factors [198]. As mentioned by reference [194], dormancy and the breaking of dormancy in buds of bamboos vary with their position on the plant, the season of the year and the species, while seed dormancy is known to occur in many tropical tree species. In seeds, several methods are known to be involved in the induction of dormancy to the germinating state. In this section, the role of plant hormones, various treatments available are discussed for bamboo seed dormancy. Important factors influencing seed germination include the seed quality and their viability. Major causes linked to the loss of seed viability are the endogenous levels of auxins and abscisic acid (ABA) during prolonged storage [199]. Besides, bamboo seeds are short-lived, germinate within 3–7 days and the germination potential is season-dependent [200]. To preserve the viability for a longer period of time, seeds are usually stored at 4 °C in desiccators with anhydrous calcium chloride. Furthermore, reference [201] revealed that prechilling the seeds (4 to 5 °C) for 4 weeks could be the most effective way to extend their life. This process is known as vernalization, and it involves exposing seeds to low temperatures in order to stimulate or to enhance seed development [198]. For instance, reference [106] stored *D. asper* seeds at 4 °C for 3 months before undergoing surface sterilization. However, degradation can occur during storage. Depending on the predominant causes of dormancy, some authors [202,203,204] have suggested various approaches to break the seed dormancy in order to improve the germination rate and speed up the germination process. Besides, the breaking of seed dormancy varies from species to species. Therefore, it is very important to determine which method and condition are the best for each plant species. Various techniques are available that enhance the vigor of seeds, and these technologies are termed as seed invigoration/seed enhancement techniques [205]. Seed invigoration is a postharvest treatment that enhances seed production by ameliorating the germinability, storability and yield performance of the seeds [199]. Hydropriming, seed hardening, on-farm priming, osmo-priming, osmo-hardening, humidification, priming with plant growth regulators, polyamines, ascorbate, salicylate, ethanol, osmolytes, coating technologies and, more recently, pre-sowing dry heat treatments are some of the treatments used to invigorate seeds [200]. These strategies provide high-value crops with value-added solutions that improve the yield and quality. Generating greater emergence rates, rapid seedling growth and better stand developmental rates are the results of seeds priming [206]. However, no treatments have been applied to *D. asper* seeds in order to break their dormancy and improve their viability. In terms of plant growth regulators, reference [207] indicated that the major gibberellins formed by the germinating embryo are GA_1_ and GA_3_. Furthermore, GA_3_ and GA_7_ are thought to activate aleurone cells, and GA_1_ and GA_4_ are thought to regulate embryo development. GA_2_ and GA_22_ are two other active gibberellins, while others like GA_12_, GA_17_ and GA_26_ show no sign of reaction. The importance of endogenous GAs as a seed germination enhancer has also been earlier emphasized by reference [208]. When the seeds of *D. membranaceus* Munro were soaked in GA_3_ solution (50 ppm) overnight, a high percent of seed germination was stimulated, with a corresponding increase in shoot length (2.70 mm) and number of sprouts (7) per explant during culture initiation [141]. Similarly, reference [209] discovered that 0.5-mg/L GA_3_ supplemented in media promotes the germination of *D. giganteus* Munro seeds under light better than BAP and Kn. In addition, GA_3_ at 50 ppm was found to be the best pre-sowing treatment on *D. hamiltonii* seeds, with a statistically significant improvement in seed viability [200]. Furthermore, seed primed with 1% KNO_3_ solution increased the germination of *D. strictus* (Roxb.) by 80.4% at the fastest rate, and no mortality was recorded when transferred to soil [210]. However, reference [211] observed that osmopriming with KCl (10%) resulted in a maximum germination percentage of 83.1% when compared to KNO_3_ and PEG-6000 on *D. strictus* seeds. Meanwhile, reference [212] soaked the *D. sinicus* seeds in 0.5% (*v/v*) potassium permanganate (KMnO_4_) for 12 h and resulted in a high germination rate.

### 6.2. Bud Position on the Bamboo Plants

The position of explants was found to affect the culture initiation and the quality of the shoots formed under in vitro conditions. During in vitro bamboo propagation, the top and bottom portions of the nodal segment in culm bamboo can hardly regenerate. The initiation of the culture is more efficient when nodal segments from a healthy mature mother plant with disinfected lateral branches are used [19]. According to reference [175], the juvenility of lateral shoots, the season of the cultures initiated and the position of axillary bud on the branch highly affect the bud break frequency in *D. longispathus*. Moreover, reference [213] reported that nodal segments from mature clumps of *B. bambos* with pre-existing axillary buds were primarily preferred as explants due to their sufficient availability all-year-round to initiate in vitro cultures, while reference [214] reported that explants from young lateral buds showed a bud break in *B. tulda*. Besides, explants from healthy mother stock were found to be good for the regeneration of new plants in *D. hamiltonii* [125], *P. stocksii* Munro [192], *G. angustifolia* and *D. giganteus* [119]. Explants taken from higher branches were found to respond better to a multiplication medium with an early bud break than explants from lower branches [94]. In *Arundinaria callosa*, the position of the nodal buds in the lateral branches affected the efficiency of the bud breaks, resulting in a higher bud break when nodal explants are taken from the basal and middle nodes compared to the distal part of the secondary branches [215]. Reference [137] illustrated that the 5th–7th positions of *B. nutans* explants from the mother stock culm were the best for the maximum regeneration in the vitro culture in bud breaking, while reference [149] found the best regeneration for *D. strictus* taken from the 1st and 2nd positions of the base of the secondary branches. Similar findings have also been reported in *D. longispathus* [175] and *B. vulgaris* [216], which mid-culm nodes of secondary branches are in the best position for axillary shoot initiation explants. Furthermore, reference [110] stated that the best explants for axillary shoot proliferation in *D. asper* were taken from the mid-culm nodes of tertiary branches. 

### 6.3. Season Collection of Explants

The period of explant collection for culture initiation was found to play an important role in reducing the level of contamination, increasing the bud break and increasing the number of shoots per explant [132]. The environmental conditions during different periods of the year varied the maturity status of the explants, hence influencing the response of explants to the culture initiation [110]. *D. asper* responded best to the culture conditions during the pre-monsoon season (May to June) but with a higher contamination rate [16], while references [19,132] stated that young branches (nodal segments) of *D. asper* collected in the spring (February–April) gave a better response in terms of lower contamination, early bud break and a higher number of shoots. On the other hand, reference [110] stated that the best time in initiating aseptic cultures for *D. asper* was in January and February, when the maximum bud break was achieved. In the spring, an increased cell division has been observed in trees as young buds produce auxins, which stimulates cell division in the cambium [217]. Moreover, the months of July–December were discovered to be unsuitable for optimal *D. asper* bud induction. Reference [19] found that during the rainy season (July–September), almost 50% of the contamination with moderate bud breaks was due to strong fungal and bacterial contaminants remaining underneath the leaf sheaths, while a poor response during the winter (October–December) was primarily due to the plant’s dormant and slow development. According to reference [110], the highest rate of contamination was also observed during the time of maximum rainfall (June–August). In a study of *B. balcooa* by reference [218], the explants collected during the rainy season in India (June–September) resulted in a high presence of contaminated explants. Furthermore, the establishment of *B. oldhamii* in vitro was a success when reference [135] collected the explant material by the end of the rainy season (June and July) in the Central-West Region of Brazil. Therefore, it is important to understand that bud break responsiveness is normally associated with the rainy season of different locations. Similar seasonal effects on bud breaks were also observed in *D. giganteus* and *B. vulgaris* [144], *B. nutans* [137,178], *B. balcoa* [162], *D. hamiltonii* Arn. Ex Munro [132] and *B. Bambos* [151].

## 7. Molecular Approaches for the Characterization of Bamboo Populations and Genomes

### 7.1. Identification and Phylogenetic Relationships of Bamboos

Traditionally, the identification of bamboo is typically focused on its morphological features, such as rhizomes, buds, leaves, branching patterns, inflorescence, flowers and fruits [219]. However, due to phenotypic plasticity and genotypic heterogeneity of the characteristics, identification can be difficult. Through genetic diversity studies, which were previously restricted by the minimal morphological characteristics, the use of molecular systems has led to significant advancements in the understanding of evolutionary relationships [220]. Research using DNA markers for breeding purposes may also discover unique alleles that appear only in one species in a population [221]. So far, several DNA-based molecular markers, such as inter-simple sequence repeats (ISSR) markers, random amplified polymorphic DNA (RAPD), restriction fragment length polymorphism (RFLP), amplified fragment length polymorphism (AFLP), single-nucleotide polymorphism (SNP), simple sequence repeat (SSR), inter-primer-binding site (iPBS) and start codon targeted (SCoT), Sequence Characterized Arbitrary Regions (SCAR) markers have been used to explore the genetic diversity of bamboo species [2,222,223]. In bamboo identification, the first report of the used (RFLP) DNA-based markers came from reference [224], after they analyzed 42 accessions belonging to six genera and about 25 species of the genus of Phyllostachys. In some earlier studies, RAPD markers were used for distinguishing species of the genera *Bambusa, Dendrocalamus, Dinocloa* and *Cephalostachyum* [225]. Two specific RAPD markers from *B. balcooa* Roxb. and *B. tulda* Roxb. were converted to SCAR in order to obtain proper markers for the molecular identification of the species [226]. Other molecular systems that were used also include SSR markers [227,228,229] for the genetic diversity studies of moso bamboo genotypes, while, recently, reference [230] emphasized the SSR markers that identify and characterize *G. chacoensis*. Twelve SSR markers were developed from this species, and among them, 10 were polymorphic, which can be used in *G. chacoensis* genetic diversity, relationships between natural populations and phylogenetics. Besides, the combined use of marker systems has been used in several studies, such as RAPD-RFLP markers, to validate the molecular categorization of *Bambusa, Dendrocalamus, Phyllostachys* and *Guadua* cultivated in Brazil [12]. They recommend RAPD-RFLP as a reproducible and informative method for screening differences among bamboo genera, species and varieties, because it is a cost-effective and effective way for bamboo conservation, management and breeding. On the other hand, reference [231] create ample EST-SSR resources that are useful for genetic diversity analyses and the molecular verification of bamboo and propose that SSR markers generated from Lei bamboo (*Phyllostachys violascens*) are more effective and reliable than ISSR, SRAP or AFLP markers.

However, there have only been a few reports on bamboo identification and phylogenetic relationships in *D. asper* using these molecular markers (Table 4). The first report to provide the essential basic genetic information of *D. asper* was conducted by reference [232]. The study showed that RAPD is a powerful technique capable of reliably differentiate closely related species using the same primers. However, due to the weak points of RAPD, AFLP was used as an alternative. According to the AFLP data, all of the specimens were in 23 distinct groups, and two groups were also distinguished from one another. To sum up, RAPD and AFLP were found to be efficient enough to reveal usable levels of DNA polymorphism and investigate the genetic information of *D. asper*. The other report that emphasized the importance of molecular applications in the determination of *Dendrocalamus* species was carried out by reference [219]. Eight species of *Dendrocalamus* present in Thailand, including *D. asper*, were employed by focusing on several PCR-based techniques, including RAPD, SCAR and multiplex PCR. From all 50 RAPD primers used, only five primers generate consistent and reproducible RAPD band patterns for all 160 individuals, while five sets of the SCAR primers were developed based on the sequences of the RAPD fragments to easily differentiate five species of *Dendrocalamus*. In addition, reference [233] used RAPD markers to classify *D. asper* genetic diversity in Indonesia. A total of 31 RAPD primers were used to produce 64 polymorphic bands, exposing the genetic diversity, and 81 distinct multi-locus genotypes were found among the 115 samples. As a result of all the studies performed on *D. asper*, the RAPD technique appears to be a valuable method for studying genetic variations in this bamboo species.

### 7.2. Clonal Fidelity

The main challenge in obtaining true-to-type germplasms in mass in vitro-produced bamboos is maintaining the elite selected characteristics, particularly in somatic embryogenesis. Somaclonal variations are a form of genetic aberrations caused by culture conditions, explant sources, ploidy levels and in vitro culture age. As stated by reference [234], exogenously applied growth regulators can disturb the cell cycle, increase the mutation rate per cell generation over time and alter DNA methylation patterns, which all can lead to the appearance of somaclonal variations. Most of these variations occur during callus induction, somatic embryo multiplication and the regeneration phase. Furthermore, since bamboo seedling populations are highly variable, using seed/seedlings for the induction of somatic embryos has more disadvantages due to their unclear genetic backgrounds. To minimize aberrations, it is recommended to use explants from mature bamboo plants rather than seeds or seedlings [235]. Moreover, callus subcultures should be limited to three times, as sustaining callus cultures for longer periods of time increases the likelihood of somaclonal variations. Therefore, determining the clonal uniformity of the in vitro-raised plants is important. An early assessment of the genetic stability of in vitro-raised plants may aid in fine-tuning the micropropagation protocol, as well as getting rid of genetically unstable plants, lowering the cost of field maintenance until maturity.

Previously, morphological, biochemical, physiological and anatomical parameters have been used to assess the genetic fidelity of in vitro-raised plants, including bamboos [19,177,234]. These morphological characteristics require extensive observations until maturity. However, due to genetic aberrations, the morphological and physiological differences can fade over the growing season, or the initially uniform-looking plants may behave differently during the flowering/fruiting stages. Furthermore, certain modifications induced during in vitro cultures may not be seen in ex vitro conditions [236]. Therefore, more effective identification tools such as DNA markers must be used to determine the genetic fidelity of in vitro-raised plants, as these markers are not influenced by the age or tissue of the plant, growth stage or environmental conditions. The same applies to genetic identification and phylogenetic relationships, several markers, viz., RFLP, RAPD, AFLP, SCARs, ISSR, SSRs, EST-SSRs and MITE-TD, have been widely applied in genetic variations among bamboos. Earlier, the detection of clonal fidelity among in vitro-raised bamboo was attempted by references [218,237], which included *B. tulda* and *B. balcoa* using RAPD markers, reference [238] screening *B. balcooa* using ISSR and reference [121] detecting clonal fidelity in B. nutans employing AFLP. The use of more than one marker system to target a wider region of the genome has been suggested by several workers. In assessing the clonal fidelity of *B. balcoa*, reference [239] reported producing 28 scorable bands from 10 ISSR primers and 61 amplicons from 21 RAPD primers. More importantly, no variation was detected among the in vitro-raised plants and the mother plant banding profiles. Similarly, references [240,241] also used two sets of markers (RAPD and ISSR) for *D. strictus* and found almost no variability among the micropropagated plantlets, indicating that the somaclonal variation was not promoted.

Despite the success of in vitro mass propagation of *D. asper*, only a few reports on the genetic fidelity of in vitro-raised *D. asper* using molecular markers have been published (Table 4). The first report to provide essential basic genetic information of *D. asper* was assessed by reference [234] on the genetic stability of in vitro raised-plants from reference [19] by analyzing the regenerated plantlets and mother plant of *D. asper* raised through enhanced axillary branching with DNA-based markers. Four sets of molecular markers were used, (RAPD, ISSR, SSR and AFLP) and did not find any somaclonal variation in the in vitro-raised plantlets. Genetic stability was confirmed (at least up to 30 passages/2 years), and the protocol developed was approved for commercial scale utilization. Reference [110] also tested the clonal fidelity of *D. asper* using RAPD and ISSR markers. The banding profiles from micropropagated plants were found to be monomorphic and similar to those of the mother plants, hence confirming their true-to-type nature. These results confirmed that *D. asper* plants obtained by axillary branching method under in vitro conditions retained their clonal fidelity. Similarly, banding profiles with ISSR markers from micropropagated plantlets were monomorphic and similar to those of the mother plants in *B. nutans* [242], *G. angustifolia* [243] and *D. hamiltonii* [244]. 

## 8. DNA Barcoding in Bamboos

Apart from DNA fingerprinting technology, DNA barcoding has been shown to have a great potential in identifying bamboo species and has recently begun to gain momentum in the taxonomical bamboo genera. No further verification of bamboo features at juvenile stage causes the absences of their discriminatory species, leading to a misidentification and mixed-up error at the nursery level. As stated by references [245,246,247], DNA barcoding is a technique that uses a short and standardized sequences of genomic DNA in discovering new or cryptic species in plants, as well as animals. Moreover, these techniques have been utilized for various purposes, including the authentication of medicinal or poisonous plants [248,249], and the assessment of biodiversity richness in a given area or community [250]. Sources from chloroplast, nuclear and multi-locus DNA are some approaches that are available in DNA barcoding techniques [30]. The mitochondrial cytochrome c oxidase 1 gene (CO1) has been exploited extensively in the Animal Kingdom and has turned out to be a suitable DNA barcode at various taxonomic levels [251,252]. However, CO1 is not an appropriate DNA marker for plant barcoding due to its slow rate of evolution [253,254]. Therefore, a number of candidate DNA markers have been evaluated for their performance as barcodes for land plants [255]. For plants, rbcL and matK are the core barcodes recommended by the Plant Working Group of the Consortium for the Barcode of Life (CBOL) [256]. Subsequent to that study, the internal transcribed spacer (ITS) or its partial sequence ITS2 were suggested as additional barcodes for plants [257,258]. Thereafter, trnH–psbA and ITS (or ITS2) were suggested as complementary markers to the core barcode of rbcL + matK [259]. Later, some studies concentrated on evaluating the identification capability of specific groups using these four barcode loci [260,261,262,263,264], and others have focused on the discovery of new markers suitable for a given taxa [265,266,267]. 

In 2012, reference [268] performed the first attempt to test the feasibility of four DNA barcoding markers (matK, rbcL, trnH–psbA and ITS) in identifying 27 species of temperate woody bamboos. Among the proposed DNA barcoding markers, three of the plastid markers showed high levels of universality, whereas the universality of ITS was comparatively low. Besides, they suggested that a combination of rbcL + ITS as a potential barcode for species discrimination. With such results, the DNA barcoding provides an alternative tool to taxonomy, especially when the diagnostic morphological characteristics are missing [269]. However, discriminating between closely related or recently evolved species or taxonomic groups with complicated evolutionary histories may challenge DNA barcoding. For instance, the recommended barcode region matK failed to discriminate against *Bambusa* species due to interspecific hybridization and polyploidy [270]. Moreover, reference [271] studied that the matK has a lower amplification potential for the bamboo species in identification. Moreover, the low discriminatory power of the core barcode (rbcL + matK) has also been reported [272]. Fortunately, however, it was found that matK barcodes for the *D. asper* showed 100% identity with four other *D. asper* accessions. Barcode obtained utilizing Maturase K primer can be utilized for identifying *D. asper* but cannot be considered as a suitable barcode for the rest of the species under study [271]. Besides, the barcode region trnH-psbA in combination with other spacer regions was ideal for discriminating the commercially important species mainly belonging to the genera *Bambusa, Dendrocalamus, Melocanna, Oxytenanthera* as well as *Ochlandra* [273]. Therefore, the aim of barcoding is to create short, standardized DNA segments in order to develop a genetic reference library for each eukaryotic individual on the planet. If developed, the specific barcode could be used as a phylogenetic identification tool for certifying planting stock materials at the nursery level before large scale plantations are established. DNA barcoding has the potential to be a novel molecular technique in complementing existing methods. 

## 9. Bamboo Genomics

Genome and transcriptome sequencing of commercially important plant species has led to the discovery of novel genes, the elucidation of biosynthetic pathways and the identification of genomic loci linked to quantitative traits. Bamboo occupies an important phylogenetic node in the grass family and the first attempt to compare the genomes of *Oryza sativa* and *Zea mays* was made by estimating the genome size of the tetraploid Moso bamboo (*Phyllostachys pubescens*) which was determined to be 2034 Mb following which, approximately 1000 genome survey sequence for the analysis of synteny [274]. Molecular markers from *O. sativa* were successfully applied and were able to resolve bamboo species into two major groups which concurred with the morphological classification as rhizome type, runner and clumper [275]. The first high quality of the draft genome of *P. heterocycla var. Pubescens* provided evidence of genome duplication and led to the identification of 31,967 genes [276]. The Bamboo genome database (Bamboo GDB) which has been developed as a direct result of multiple genome sequencing projects now provides researchers with a library of functionally annotated genes and pathways as well as tools for analysis and graphical representation of data sets [277]. Since then, transcriptome analysis of *P. edulis* has led to the discovery of genes linked to floral transition and flower development in bamboo, both of which are pertinent to the breeding industry [278]. The cumulative data provides an important resource for the development of molecular markers for the characterization of genome variation in bamboo via genome resequencing [279]. The wealth of information related to microsatellites has facilitated the reconstruction of high-resolution phylogenetic maps of bamboos [227]. The discovery of transposable genetic elements within the bamboo genome, which are responsible for somaclonal variation, has also provided insights into the phylogeny of Asian bamboos [280]. Transcription factors are important for the regulation of genes and their role in growth and development makes them of importance to genetic engineering, the characterization of these transcription factors in *P. edulis* has provided the foundation for the discovery and application of novel transcription factors for downstream applications in genetic modification [280]. The recent publication of the draft genome sequence of the diploid, herbaceous bamboo *Raddia distichophylla* (Schrad. ex Nees) Chase, has provide a clearer understanding of the process of lignification and the genes associated with this biosynthetic pathway [281]. The increase in the availability of both genome sequencing data from multiple projects when integrated with transcriptomic data from different developmental stages [282,283] will provide researchers and commercial breeders with data that can be applied for the improvement of bamboo via the application of Marker Assisted Breeding (MAS) program and genetic engineering of important regulatory pathways.

## 10. Summary and Future Outlook

Two of the key factors for sustainable development in the bamboo industry are the conservation of their germplasm and improving the genotype. With the realization that in vitro propagation technology is essential in meeting the ever-increasing demands of planting stock, micropropagation is the best available technique and will become the standard for the commercial-scale propagation of bamboo in the future. Bamboo tissue culture has the advantages of having a greater number of genotypes in the culture, from which propagation may precede to ensure a greater diversity of the species. Additional efforts and research of all the possible factors are required for standardizing the micropropagation techniques in a majority of the important bamboo species so that they can be used for improvement and clonal forestry programs, as well as for ex situ conservation and cryopreservation of the species. Furthermore, the establishment of callus cultures and establishment of plants regenerated from calli or cell suspensions via somatic embryogenesis hold the potential to produce novel soma-clonal variants of bamboo. Questions are often raised as to the genetic fidelity of the in vitro-raised plants, especially those raised through somatic embryogenesis. Therefore, the biotechnological interventions of DNA fingerprinting and DNA barcoding have increased in many branches and subdisciplines of biology as a tool for identifying DNA polymorphisms between individuals, studying genetic diversity and analyzing the genetic distance between species, as well as ascertaining their genetic fidelity.

Among the fast-growing woody bamboos of the world, *D. asper* adorns a special position as one of the versatile species for industrial applications. Having a slow pace of breeding is one of the challenges faced in most bamboo species, including *D. asper*, which is primarily associated with long intervals and large population sizes. Transgenic breeding through in vitro regeneration techniques for the genetic improvement of bamboo is one of the biotechnological approaches that needs to be improved, as tissue culture technology can be used for rescuing the hybrid seeds produced by conventional breeding methods. Besides, using the genetic base of biotechnology tools, including in vitro selection, mutagenesis and transgenics, can be applied to assist in developing a controlled flowering condition for bamboo breeding. Toward achieving this, the development of germplasm resources is essentially required, which is still lacking in the *D. asper* database system. In bamboo, only a few reports are available regarding breeding developments. Further inputs are required for the establishment of an efficient suspension cell culture system and development of a transformation and transgenics procedure with high efficiency in bamboos. Therefore, an efficient tissue regeneration system needs to be developed in *D. asper* for taking advantage of genetic transformation and gene editing for improvement. The transcription factors (TFs) in regulating the gene expression are important in the development of genomic resources for identifying genes key traits. These TFs are DNA-binding proteins that control gene transcription in order to regulate gene expression. TFs regulate several ontogenetic responses in bamboo, especially those that are environmentally triggered, and play a significant role in the stress response in bamboo. Moreover, another critical regulatory element in higher plant genomes is noncoding RNAs. These RNAs are small but powerful in regulating plant growth and development. However, to the best of our knowledge, there are still no details of the transcription factors (TFs) identified in *D. asper*, as well as reports on noncoding RNAs. Besides, there is still much more unknown information on the whole genome of *D. asper* biology. However, the availability of transcriptomic, proteomic and metabolomic technologies are the routes in collecting genomic resources that will bring advantages to bamboo development that underlie the phenotypic and ecological divergence of bamboo.

Owing to their perennial nature, bamboos encounter stress factors such as cold, drought, salinity and high temperature more commonly than any other grass species. Currently, natural tolerance is the only source available in combating these adverse factors. Along with the cloning of promising genotypes selected from the natural habitat, efforts are required to generate variations and to screen useful traits, particularly for stress tolerance. As there is a conspicuous absence of a strong breeding system, breeding toward stress tolerance remains underdeveloped in *D. asper*. Various types of genes that contribute to this stress tolerance have been identified in several types of bamboo. However, there are no reports found in *D. asper*. Knowing the expression of the genes, the adaptation mechanism to counter stress effects, altering the phenotypic features, the details of the network of genes involved in stress response and their relationship to development growth need to be explored further. The latest advancements in gene editing technologies have found increasing uses of clustered regularly interspaced short palindromic repeat (CRISPR)–CRISPR-associated endo-nuclease (Cas9) systems and nanopore sequencing technology. By targeting the alleles of a gene, this technology could demonstrate the plant change genes in bamboo. Moreover, selected studies have shown the utilization of the CRISPR-Cas9 system in developing plant varieties that can tolerate adverse environmental conditions. Therefore, this technology awaits the application of *D. asper* research in the gene-editing approaches, especially in combating the stress tolerance, as well as climatic change, faced by the species. 

## Figures and Tables

**Figure 1 plants-10-01897-f001:**
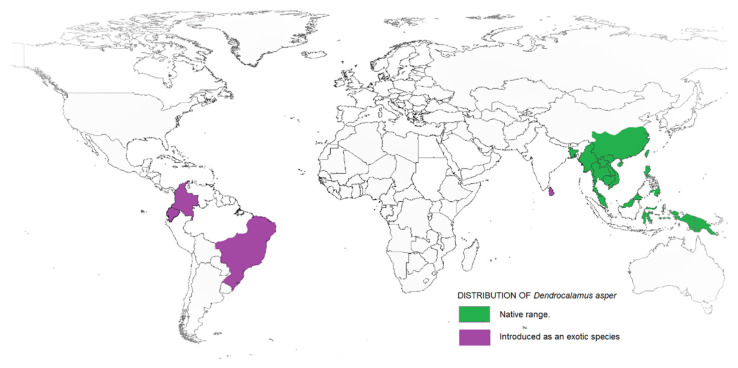
Range and distribution of *D. asper* in its native and introduced habitats.

**Figure 2 plants-10-01897-f002:**
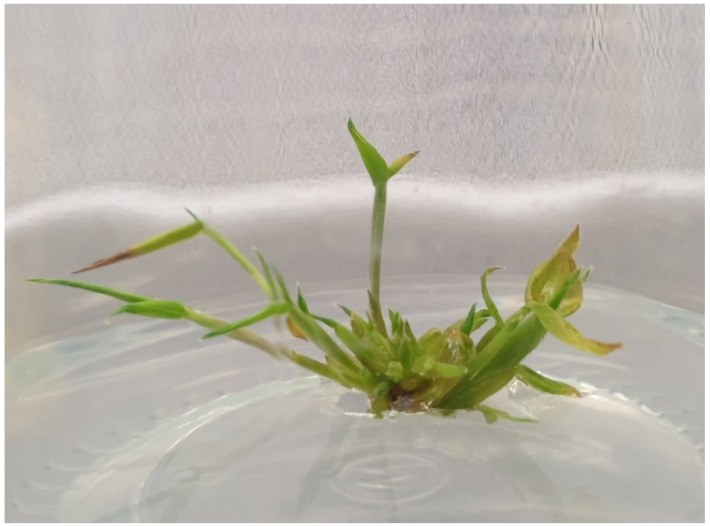
Micropropagation of *D. asper* using a seeds explant. Shoot multiplication in Murashige and Skoog basic medium (MS) + 2 mg/L BAP.

**Table 1 plants-10-01897-t001:** Successful micropropagation of *D. asper* via organogenesis from various explants.

Explant	Basal Medium	PGRs (as Indicated in μM Except Otherwise Mentioned)	Results	Reference(s)
Node (young first three segment)	MS	* BAP (22.0) ** BAP (22.0) + AdS (216.0) *** IBA (4.90)	Shoot multiplication and rooting	[16]
Node	MS	* BAP (15.0) ** BAP (10.0) + AdS (75.0) *** ½ MS + IBA (5.0) + NAA (5.0)	Shoot multiplication and rooting	[19]
Seeds, Nodes	MS	* BAP (13.32) ** BAP(13.32) ***NAA (16.11); IBA (49.0)	Shoot multiplication and rooting	[91]
Node	MS	** BAP (31.08) ***NAA (16.11) + IAA (5.71)	Organogenesis, multiple shoots and rooting	[92]
Nodes	MS	** BAP (13.32) + Ads (270.0) *** IBA (4.90)	Shoot multiplication and rooting	[93]
Node	MS	* BAP (8.86) ** BAP (8.86) + Ads (13.5) + 3% Suc ***IBA (14.76) + NAA (3.67) + 3% Suc # 2,4-D (14.61) ##, ** 2,4-D (14.61) *** IBA (14.76) + NAA (3.67)	Shoot multiplication and rooting	[94,100]
Node	MS	* ¼ MS BAP ** ¾ MS + 3 ppm Kn	Shoot multiplication	[96]
Stem cuttings	MS	* BAP (0–8.88) + CW (0–20.0) ** BAP (22.2) *** NAA (2.68) + AA (283.5) + CA (130) + Cyst (206.25)	Shoot multiplication and rooting	[97]
Small branch cuttings	MS	** BA (3 × 10^−5^)	Shoot multiplication	[98]
Inter node segments	MS	* BAP (2.22) ** BAP (8.88) # Kin (23.25) + NAA (16.11)	Shoot multiplication, rooting and callusing	[104]
Clump	MS	* BAP (15) ** mT (20)	Shoot multiplication	[101]
Immature and mature inflorescence	MS	* BAP (31.08) ** BAP (13.32) *** IBA (49.0)	Shoot multiplication and rooting	[102]
Seeds	MS	* BAP (22.2) ** BAP(1.332) * IBA (49.0) + NAA (16.11)	Shoot multiplication and rooting	[105]
Seeds	MS	* BAP (22.2) ** BAP (13.32) *** IBA (49.0); NAA (16.11)	Shoot multiplication and rooting	[106]
Seeds	MS	* BA (20.0) ** BA (10.0) *** IBA (40.0)	Shoot multiplication and rooting	[107]
Node	MS	* TDZ (1.135) + NAA (1.34) + AA (283.5) + CA (130.0) + Cyst (206.25) ** TDZ (1.135) + NAA (1.34) + AA (283.5) + CA (130.0) + Cyst (206.25) *** ¼ MS + IBA (9.80)	Shoot multiplication and rooting	[108]
In vitro grown shoots	MS	** BAP (31.08) • NAA (16.11) + IBA (14.70) + 5% Suc	Shoot multiplication and rhizogenesis	[109]
Seeds	MS	* BAP (13.32) ** BAP (13.32) *** IBA (34.30)	Shoot multiplication and rooting	[110]

AA, ascorbic acid; AdS, adenine sulphate; BAP, 6-Benzylaminopurine; CA, citric acid; CW, coconut water (milk); Cyst, cysteine; IAA, indole-3-acetic acid; IBA, indole-3-butyric acid; Kin, kinetin; MS, Murashige & Skoog medium; mT, Meta-topolin 6-(3-hydroxybenzylamino) purine; NAA, α-naphthaleneacetic acid; PGR, plant growth regulator; Suc, sucrose; TDZ, thidiazuron. *, Seed germination/shoot induction; **, Shoot proliferation/multiplication; ***, rooting; •, Rhizogenesis; #, callus initiation; ##, callus proliferation.

**Table 2 plants-10-01897-t002:** The successful micropropagation of *D. asper* through somatic embryogenesis.

Explant	Basal Medium	PGRs (as Indicated in μM Except Otherwise Mentioned)	Results	Reference(s)
Node	MS	# MMS + 2,4-D (30.0) * BAP (20.0) *** NAA (5.0–25.0)	Somatic embryogenesis and germination	[95]
Nodal and leaf bases	MS	# 2,4-D (30.0) ## 2,4-D (9) + IAA (2.85) + BAP (0.88) X BAP (4.4) + GA_3_ (2.8) ** BAP (13.2) *** NAA (16)	Somatic embryogenesis	[111]
Seeds	MS	#,## 2,4-D (13.59) or 2,4-D (2.265) *,** BAP (8.88) + NAA (2.68) + Kin (4.65) *** ½ MS + IBA (13.32)	Somatic embryogenesis	[112]

2,4-D, 2,4-dichlorophenoxyacetic acid; BAP, 6-Benzylaminopurine; GA, Gibberellic acid; IAA, indole-3-acetic acid; IBA, indole-3-butyric acid; Kin, kinetin; MS, Murashige & Skoog medium; MMS, Modified Murashige & Skoog medium; NAA, α-naphthaleneacetic acid; PGR, plant growth regulator; #, initiation of embryogenic callus; ##, embryogenic callus proliferation; X, plantlet development from somatic embryos *, Somatic embryo shoot induction; **, somatic embryo shoot proliferation/multiplication; ***, rooting.

**Table 3 plants-10-01897-t003:** List of sterilants used for different explants in *D. asper* disinfection.

Explant	Sterilant Used	Concentration (%)	Duration (Minute)	Reference(s)
Nodal segment	Teepol	A few drops	5	[16]
	Bavistin	1	30	
	Mercuric chloride	0.1	6–8	
Nodal segment	Teepol	3	10	[19]
	Streptomycin sulphate	0.2	10	
	Tetracycline hydrochloride	0.2	10	
	Bavistin	0.2	15	
	Mercuric chloride	0.1	3–5	
	Ethyl ethanol	70	1	
Nodal segment	Mercuric chloride	0.1	7–10	[91]
	Cetavelon	5	15	
Nodal segment	Bavistin	0.2	20	[93]
	Tween 20	A few drops	20	
	Mercuric chloride	0.1	20	
Immature inflorescences	Ethyl ethanol	70	1	[102]
	Mercuric chloride	0.1	8–15	
Seeds	Sodium hypochlorite	4	20	[106]
Nodal segment	Bavistin	0.1	20	[104]
	Streptomycin sulphate	0.04	20	
	Ethyl ethanol	70	1	
	Mercuric chloride	0.04	6	

**Table 4 plants-10-01897-t004:** The details of genetic diversity studies and output of *D. asper* using various molecular bamboos.

Tools Used	Achievements	Reference(s)
	Genetic diversity studies	
RAPD, SCAR and multiplex PCR	The analysis and screening for the polymorphism Dendrocalamus bamboos using the SCAR primer was rapid and efficient.	[219]
RAPD and AFLP	The first report to provide essential basic genetic information of *D. asper*. RAPD and AFLP were found to be efficient enough to reveal usable levels of DNA polymorphisms and to investigate genetic information.	[232]
RAPD	Using 31 RAPD primers producing 64 polymorphic bands.	[233]
	Genetic fidelity testing of in vitro-raised *D. asper*.	
Morphological descriptors	Compared in vitro-raised plants with mother plants and found no variation.	[19]
RAPD, ISSR, SSR and AFLP	Tested clonal fidelity of propagated in vitro for over 2 years (at least 30 passages) via enhanced axillary branching with mother plant and found no somaclonal variation.	[234]
RAPD and ISSR	DNA samples from in vitro-grown shoots under various stages of subculture, hardened plants growing in the greenhouse, plants growing in the field and the mother plant were subjected for clonal fidelity. No polymorphism was found, confirming true to type nature in vitro raised plants.	[243]

## Data Availability

Not Applicable.

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
