# Peer review of "A Concise Review of Dendrocalamus asper and Related Bamboos: Germplasm Conservation, Propagation and Molecular Biology"

_plants, 2021, doi:10.3390/plants10091897_

Round 1

Reviewer 1 Report

The review article entitled - A Concise Review of Dendrocalamus asper: Germplasm Conservation and Propagation is written and gives an overview of one of the most important aspects of Dendrocalamus asper micropropagation and the molecular approaches. I would like to recommend this article for acceptance for publication after the minor corrections.

Minor corrections -

Once started using the abbreviated form, please continue with it. I could see, for example, Dendrocalamus asper repeated after the abbreviated form (D. asper). Follow italics for the scientific names. Few places, the words are misspelled.

Introduction

1. Having 120 genera and 1642 species [3] .... As per Canavan et al., 2017, there are 1662 species in 121 genera (Susan Canavan, David M. Richardson, Vernon Visser, Johannes J. Le Roux, Maria S. Vorontsova, John R. U. Wilson, The global distribution of bamboos: assessing correlates of introduction and invasion, AoB PLANTS, Volume 9, Issue 1, January 2017, plw078, https://doi.org/10.1093/aobpla/plw078). Kindly provide the latest information after having a close look.
2. Reference 5 looks very old. Please, provide the latest reference claiming the distribution pattern.

2. Dendrocalamus asper

3. Bambusa, Cephalostychum, Dendrocalamus, Gigantochloa, melocanna, are reported....
Please follow a common pattern of writing the genera name. italicize or not.

5.3. Hardening and Acclimatization
4. By adapting the micropropagation after micropropagation.... Please check

5.5. Sterilants
5. (NaOCl) was also employed by different... Please check

5.6. Selection of Antibiotic and Fungicide
6.According to a by [103],.... Change to According to [103],....

Author Response

Dear Professor,

Thank you very much for your kind review. The responses are attached.

Best regards,

Authors.

Reviewer 2 Report

Please read the cover letter

Author Response

(The authors gave the same response as above.)

Reviewer 3 Report

The manuscript is a detailed study of the propagation of the giant bamboo (Dendrocalamus asper). Some suggestions for improving the manuscript are included.

Section 5 - information about other factors affecting micropropagation should be added: the composition of the medium (e.g., the concentration and type of carbohydrates), the cultivation conditions (e.g., light intensity)

Section 7 - information on genome sequencing of bamboo species should be added, as it is useful for the application of molecular markers

Tables 1 and 2 – BA (6-benzyladenine) and BAP (6-benzylaminopurine) are synonyms; one of the two names should be left in the text

Table 2 shows research on somatic embryogenesis, but ref. 82, 84, 90 and 100 refer to in vitro propagation through organogenesis

Page 8, 8th line from bottom - Pinus roxburghii [105] is pine, not bamboo

References - a list of journal abbreviations should be checked, cited journals should be abbreviated according to ISO 4 rule.

Author Response

(The authors gave the same response as above.)
